# Advancements in Autophagy Modulation for the Management of Oral Disease: A Focus on Drug Targets and Therapeutics

**DOI:** 10.3390/biomedicines12112645

**Published:** 2024-11-19

**Authors:** Md Ataur Rahman, Mushfiq Hassan Shaikh, Rajat Das Gupta, Nazeeba Siddika, Muhammad Saad Shaikh, Muhammad Sohail Zafar, Bonglee Kim, Ehsanul Hoque Apu

**Affiliations:** 1Department of Oncology, Karmanos Cancer Institute, Wayne State University, Detroit, MI 48201, USA; 2Department of Otolaryngology-Head and Neck Surgery, Western University, London, ON N6A 4V2, Canada; mushfiq.shaikh@lhsc.on.ca; 3School of Medicine, University of Texas Rio Grande Valley, Edinburg, TX 78539, USA; 4Department of Epidemiology and Biostatistics, Arnold School of Public Health, University of South Carolina, Columbia, SC 29208, USA; rajat89.dasgupta@gmail.com; 5Oral Health Sciences Division, College of Dental Medicine, Lincoln Memorial University, Knoxville, TN 37923, USA; nazeeba.siddika@lmunet.edu; 6Department of Oral Biology, Sindh Institute of Oral Health Sciences, Jinnah Sindh Medical University, Karachi 75510, Pakistan; drsaadtanvir@gmail.com; 7Department of Clinical Sciences, College of Dentistry, Ajman University, Ajman P.O. Box 346, United Arab Emirates; drsohail_78@hotmail.com; 8Centre of Medical and Bio-Allied Health Sciences Research, Ajman University, Ajman P.O. Box 346, United Arab Emirates; 9School of Dentistry, Jordan University, Amman 19328, Jordan; 10Department of Dental Materials, Islamic International Dental College, Riphah International University, Islamabad 44000, Pakistan; 11Department of Pathology, College of Korean Medicine, Kyung Hee University, 1-5 Hoegidong Dongdaemun-gu, Seoul 02447, Republic of Korea; bongleekim@khu.ac.kr; 12Korean Medicine-Based Drug Repositioning Cancer Research Center, College of Korean Medicine, Kyung Hee University, Seoul 02447, Republic of Korea; 13Department of Biomedical Sciences, College of Dental Medicine, Lincoln Memorial University, Knoxville, TN 37923, USA; 14Centre for International Public Health and Environmental Research, Bangladesh (CIPHER,B), Dhaka 1207, Bangladesh; 15Department of Internal Medicine, Division of Hematology and Oncology, University of Michigan, Ann Arbor, MI 48105, USA

**Keywords:** autophagy, molecular biology, oral health, oral cancer, periodontitis, therapeutic target, preventive mechanism

## Abstract

Autophagy is an intrinsic breakdown system that recycles organelles and macromolecules, which influences metabolic pathways, differentiation, and thereby cell survival. Oral health is an essential component of integrated well-being, and it is critical for developing therapeutic interventions to understand the molecular mechanisms underlying the maintenance of oral homeostasis. However, because of the complex dynamic relationship between autophagy and oral health, associated treatment modalities have not yet been well elucidated. Determining how autophagy affects oral health at the molecular level may enhance the understanding of prevention and treatment of targeted oral diseases. At the molecular level, hard and soft oral tissues develop because of complex interactions between epithelial and mesenchymal cells. Aging contributes to the progression of various oral disorders including periodontitis, oral cancer, and periapical lesions during aging. Autophagy levels decrease with age, thus indicating a possible association between autophagy and oral disorders with aging. In this review, we critically review various aspects of autophagy and their significance in the context of various oral diseases including oral cancer, periapical lesions, periodontal conditions, and candidiasis. A better understanding of autophagy and its underlying mechanisms can guide us to develop new preventative and therapeutic strategies for the management of oral diseases.

## 1. Introduction

Autophagy is a highly regulated catabolic process that mediates cellular destruction and is essential for the preservation of homeostasis and development of tissues [1]. Its primary function is to eliminate damaged and unnecessary organelles, misfolded proteins, and pathogens that have invaded the cell [2]. Recently, a dual role of autophagy has been identified for various oral disorders [3]. For example, autophagy has the potential to prevent and promote growth of odontogenic tumors [4]. The development of oral disease involves various molecular and cellular pathways, including PI3K/AKT/mTOR, NF-κB, Ras-Raf-MEK-ERK, Wnt signaling, and Hippo pathways [5]. The early stage involves the induction of autophagy as a survival mechanism. In the later stage, the Akt/mTOR/survivin signaling pathway reduces the autophagy [6]. This is further complicated by complex dynamic oral environmental factors such as frequent introduction of oral microorganisms, high metabolic need for oral mucosa cell turnover, and cyclic mechanical stresses on oral tissues [7]. Apart from these pro-survival benefits, autophagy is known for type II programmed cell death because of powerful pro-autophagic stimulation [8]. Similarly, autophagy plays a role in other oral physiological processes such as the growth of teeth and the relationship with oral microbial flora [9]. During aging, rising levels of reactive oxygen species (ROS) cause a reduction in the enamel organic matrix, which results in increased crystallinity and rigidity than younger enamel [10]. Although these senile changes in enamel reduce the risk of developing caries, the tooth becomes more fragile and prone to fracture in case of physical trauma. Thus, the impact of autophagy on oral disorders is variable and often depends on levels and disease progression.

Understanding the mechanism and effects of autophagy at the molecular level may guide the prevention and treatment of oral diseases [9]. However, further research is needed to fully understand the role of autophagy in the progression of oral health and to establish an autophagy treatment approach for oral health. The significant advancements in autophagy research have provided fresh insights into the pathophysiology of oral disorders, particularly oral cancer, periodontal diseases, periapical lesions, and oral candidiasis. This article critically reviews various aspects of autophagy and their significance for various oral diseases including oral cancer, periapical lesions, periodontal conditions, and candidiasis. In addition, therapeutic approaches for autophagy modulation and management of oral diseases are discussed.

## 2. Relationship Between Autophagy and Oral Health

Autophagy serves as a critical mechanism for maintaining cellular homeostasis by facilitating the degradation of intracellular materials, including macromolecules and organelles [11]. Basal autophagy operates in all cells and is upregulated in response to various stress stimuli to mitigate damage. Autophagy involves the formation of double-membrane vesicles called autophagosomes, which encapsulate targeted material and subsequently fuse with lysosomes to form autolysosomes. The lysosomal environment characterized by low pH and hydrolytic enzymes facilitates the degradation of sequestered materials [12,13]. Under physiological conditions, autophagy functions at basal levels, clearing damaged organelles, misfolded proteins, and pathogens, thereby acting as an endogenous defense mechanism during adverse conditions such as hypoxia, heat, starvation, and oxidative stress [12,14,15]. In addition, autophagy exhibits dual roles in cell fate, acting as a pro-survival mechanism under certain conditions and contributing to a pro-death process termed autophagic cell death in specific contexts [16]. The overarching term “autophagy” encompasses three primary types: macro-autophagy, micro-autophagy, and chaperone-mediated autophagy (CMA), each differing in their delivery patterns and physiological functions [17]. For example, CMA relies on the recognition of KFERQ-motif bearing proteins by cytosolic heat shock protein A8 (HSPA8) and subsequent translocation into lysosomes through lysosomal-associated membrane protein 2A (LAMP2A) multimerization [18]. In macro-autophagy, phagophores sequester random cytoplasmic cellular components, damaged organelles, and microbes, expanding into autophagosomes through lipid acquisition [19]. In terms of pathological conditions, autophagy intricately influences tumor suppression, promotion, inflammation, and immune responses [20,21,22]. The mechanistic steps of autophagy involve four key stages: (a) induction and cargo packaging, (b) elongation of the phagophore, (c) autophagosome formation and completion, and (d) lysosomal fusion and breakdown, with the autophagy-related genes (ATGs) serving as central regulators [19,23].

The intricate regulation of autophagy emphasizes its multifaceted role in cellular processes and its responsiveness to various signaling pathways (Figure 1). Central to autophagy is the role of ATGs, with Microtubule Associated Protein 1 Light Chain 3 (LC3 or MAP1LC3) playing a crucial role [24]. LC3, cleaved by ATG4, forms LC3-I, which upon conjugation with phosphatidylethanolamine by the ATG5-ATG7 complex transforms into LC3-II. LC3-II binds to autophagosome membranes, facilitating their elongation, and serves as a key biomarker for detecting autophagy [24,25,26]. In addition to the formation of autophagosomes, the autophagy flux, which signifies the progression from sequestration to degradation of cargo within lysosomes, is essential. Chemical inhibitors like bafilomycin-A1 and chloroquine block autophagosome–lysosome fusion and hinder autophagy, leading to the accumulation of autophagosomes. Careful consideration of multiple markers, such as Sequestrosome1 (SQSTM1) or p62, is crucial for accurate autophagy assessment [27,28]. Furthermore, autophagy is finely regulated by stress signaling pathways, notably the ATG1 or Unc-51-like autophagy-activating kinase (ULK1) switch activated by AMP-activated protein kinase (AMPK) and repressed by mechanistic target of rapamycin (mTOR), a protein kinase. AMPK activation and mTOR repression, often induced by starvation, are common autophagy regulators [29,30]. ATG1 activation phosphorylates and activates Beclin 1 (BECN1) (a membrane-transforming protein), initiating the formation of autophagosomes through the catalytic activity of phosphatidylinositol 3-phosphate kinase (PtdIns3K) complex [31,32]. Inflammatory pathways, particularly those converging on NFκB, also have shown to modulate autophagy. NFκB activation, dependent on IκB kinase (IKK) complex-mediated degradation of IκB proteins, induces autophagy-related gene expression, enhancing autophagy. However, NFκB may also attenuate autophagy by promoting mTOR pathway component expression [33,34,35].

Oral health is a multifaceted concept encompassing the condition of the mouth, teeth, and orofacial structures, and is crucial for fundamental functions such as mastication, breathing, and speaking [36]. It also embodies the absence of persistent orofacial pain, occurrences of oral and pharyngeal cancer, oral tissue abnormalities, and congenital anomalies such as cleft lip and palate [37]. Beyond the physiological aspects, oral health extends into psychosocial dimensions, influencing self-confidence, overall well-being, and the capacity to engage in social and occupational activities without pain, discomfort, or embarrassment. It is integral to the overall well-being of individuals, encompassing the physiological, pathological, and psychosocial aspects of the oral cavity [38]. The oral cavity is a dynamic environment comprising hard and soft tissues, including teeth, gums, oral mucosa, salivary glands, and supporting structures. Teeth, essential for mastication and speech, are composed of enamel, dentin, and pulp, and their health is crucial for overall oral function. Gingival tissues and the periodontium play a pivotal role in tooth support and protection against microbial infiltration. Saliva is a complex biofluid that contributes to oral homeostasis, aids in digestion, speech, antimicrobial defense, and tissue lubrication. A spectrum of diseases and conditions, such as dental caries, periodontal diseases, oro-dental trauma, oral cancer, malocclusions, and congenital anomalies, contributes to the complexity of the oral environment [39,40,41] (Figure 2). In addition, oral health is intricately linked to systematic conditions such as diabetes, bacterial endocarditis, cardiovascular diseases, hypertension, stroke, and respiratory disorders [42,43,44]. Therefore, maintaining effective oral health is of fundamental importance to maintain overall systemic health.

According to Yu et al. (2017), *Fusobacterium nucleatum* promotes colorectal cancer (CRC) by altering autophagy; it was found that *F. nucleatum* is abundant in CRC tissue and enhances autophagy to withstand chemotherapies. This bacterium activated autophagy by upregulating miR-18a* and downregulating ATG7 and ATG16L1, which are autophagy-related genes, via the TLR4 and MYD88 signaling pathways. In instances with *F. nucleatum*, blocking autophagy may enhance CRC therapy outcomes by reducing cancer cell survival. Depressing autophagy may increase CRC cell susceptibility to chemotherapeutic drugs, overcoming resistance and improving treatment success [45]. Another study Zhang et al. (2015) found that autophagy induction causes ovarian cancer paclitaxel resistance. Paclitaxel therapy activated autophagy in ovarian cancer cells, which removes damaged organelles and proteins to promote cell survival under stress. Ovarian cancer cells were more sensitive to paclitaxel when autophagy was suppressed, suggesting that autophagy protects cancer cells from cell death [46].

The oral cavity, as the initial segment of the digestive tract, plays a crucial role in food processing, housing the oral mucosa composed of non-keratinized stratified squamous epithelia [47]. Unlike keratinocytes, oral epithelial cells are directly exposed to saliva, water, and food remnants, and are subject to varied environments derived by dietary intake [48]. The oral epithelial cells, when exposed to certain stress stimuli such as tobacco smoking, alcohol, or human papillomavirus (HPV) demonstrated increased autophagic activity as a defensive mechanism [49,50]. Moreover, the oral cavity is characterized by continuous remodeling and regeneration; hence, it relies heavily on autophagy [51]. This significance is further emphasized by the oral cavity’s susceptibility to bacterial colonization, with numerous oral diseases stemming from bacterial infections and ensuing immune responses [51]. Typically, autophagic tone declines with age; however, increases in conditions such as periodontitis, oral cancer, chronic oral infections, and dental senescence occur [23]. Autophagy also plays a pivotal role in managing infectious agents, including intracellular pathogens, while also exerting control over inflammatory pathways, influencing myeloid/lymphoid cell differentiation, and orchestrating multicellular immunity [51].

The delicate balance of autophagy is crucial for maintaining cellular homeostasis in the oral cavity. Autophagy intricately contributes to various aspects of oral tissue renewal across different anatomical sites within the oral cavity, such as the gingiva, oral mucosa (including tongue), periodontal tissue, salivary glands, and dental pulp. Gingiva is a crucial component of the oral cavity, and benefits from autophagy’s ability to maintain cellular homeostasis and eliminate damaged components. Research suggests that autophagy contributes to gingival fibroblast survival and function, emphasizing its importance in the regeneration of gingival tissues [52]. In the case of the periodontium, comprising the periodontal ligament and alveolar bone that undergoes constant remodeling and regeneration, autophagy plays a vital role in maintaining periodontal tissue homeostasis, influencing osteoblast and osteoclast activities. This finely tuned autophagic regulation contributes to the equilibrium required for periodontal tissue integrity [53,54]. Similarly, within the dental pulp, autophagy emerges as a key player in cellular responses to stress and inflammation. It aids in preserving the vitality of dental pulp cells and supports reparative processes. Autophagy’s involvement in the removal of damaged organelles contributes to an environment favorable for dental pulp to regenerate [55,56].

Autophagy also plays a role in salivary gland regeneration, influencing both acinar and ductal cells. By facilitating the removal of dysfunctional cellular components, autophagy supports salivary gland homeostasis. This dynamic process is particularly relevant in the context of maintaining oral health and function [57,58]. Certain inflammatory oral conditions, such as periodontitis and oral mucositis, are intricately linked with compromised tissue integrity. The dysregulation of autophagy can contribute to the pathogenesis of these diseases [59]. The understanding of interactions between autophagy and inflammation is crucial for devising therapeutic strategies for treating oral diseases [9]. Furthermore, autophagy emerges as a positive regulator of oral mucosa repair and regeneration, as oral mucosa possess robust regenerative capabilities due to abundant vascularization and unique repair mechanisms [60]. However, the extent of autophagy significantly influences the outcomes of oral mucosa repair. An earlier study revealed that activated autophagy enhances the expression of α-smooth muscle actin (αSMA) and type 1 collagen production in oral mucosa, leading to fibrotic repair [60]. However, attached gingiva shows no autophagy activation during wound healing, resulting in a scarless restoration outcome without extra collagen deposition or myofibroblast differentiation [60]. Autophagy also plays a role in pathological disorders, for instance hypertonicity, oral submucous fibrosis, and carcinogenesis. Arecoline-induced overactivated autophagy in oral submucous fibrosis leads to apoptosis and dysregulation in oral keratinocytes [61]. Recent studies demonstrated that autophagy suppression alleviates fibrosis by promoting apoptosis and inhibiting fibroblast proliferation, and increased autophagy levels during oral carcinogenesis correlate with poorer outcomes in leukoplakia [62]. Autophagy’s role in oral cancer angiogenesis is context dependent and varies with cancer type and malignancy stage [63]. Autophagy helps tumor cells overcome nutrient deficit and enhances survival at the premature stage but suppresses tumor promotion through the activity of ROS and mTOR signaling pathways after tumor growth [63,64]. Ultimately, autophagy emerges as a crucial modulator in influencing diverse aspects of oral health from periodontitis, wound healing, and angiogenesis to the complex interplay in oral cancer progression. The intricate involvement of autophagy in maintaining cellular homeostasis, supporting reparative processes, and modulating inflammatory responses underscores its significance in the broader landscape for oral health.

## 3. Interplay Between Autophagy and Oral Diseases

The interplay between autophagy and oral diseases reveals a complex link that affects oral cavity disorders’ progression, severity, and treatment effects. Exploring the various functions of autophagy in oral health and disorders provides insights into its significance for medical treatment and therapeutic strategies. Oral diseases refer to a range of problems that impact the oral cavity including highly prevalent dental caries, periodontal diseases, and oral cancer [65]. Autophagy plays a significant role in preserving oral health by promoting removal of cellular components and controlling inflammatory reactions [66]. Autophagy, in the context of dental caries, aids in eliminating damaged cellular waste and reducing bacterial invasion, hence serving as a protective mechanism against tooth decay [67]. On the other hand, if autophagy is not well regulated, it can weaken the integrity of oral tissues, making individuals more susceptible to various oral infections and inflammatory disorders.

Periodontal disorders mostly arise from infections and inflammation of the gums, periodontal ligaments, and alveolar bone. During the initial phase known as gingivitis, the gums may exhibit swelling, redness, and bleeding [68]. Periodontitis is a major public health issue worldwide resulting in the degeneration of periodontal tissues [69]. Recent findings indicated a two-way connection between autophagy and periodontitis, with autophagy playing a role in regulating inflammatory responses and affecting the balance of periodontium [70]. Oral keratinocytes, dendritic cells, and macrophage autophagy facilitate the spread of *Porphyromonas gingivalis* (*P. gingivalis*) [71]. After entering myeloid dendritic cells, *P. gingivalis* and RAB5A-positive vesicles co-localize in early endosomes. *P. gingivalis* virulence factors increase BECN1 and LC3-II and decrease annexin V staining and caspase activation in eukaryotic cells, suppressing apoptosis and forming autophagosomes [72] (Figure 3). Compared to tetra-acylated LPS, penta-acylated LPS from *P. gingivalis* increases LC3 vesicle formation and quadruples autophagosome diameter. *P. gingivalis*-dependent autophagosome formation requires PAMP-TLR interaction, since TLR4 detects penta-acylated LPS but not tetra-acylated [73]. Melanoregulin (MREG), a protein essential for β-N-Acetylglucosaminidase lysosomal hydrolase and Cathepsin D activity, was lowered by this penta-acylated LPS from *P. gingivalis*, showing that pathogen-associated molecular patterns (PAMPs) degrade lyso oral mucosal cells’ C-type lectin receptor. CD209/DC-SIGN recognizes *P. gingivalis*’ fimbria major subunit Mfa1’s mannose content, downregulating Lamp1 expression [74]. *P. gingivalis* hides in autophagosomes to collect nutrients for reproduction; however, it hinders lysosome fusion to escape disintegration [71]. *P. gingivalis*’ increased autophagosome production is misconstrued as lower autophagy flux [75]. Calcitriol, 1α 25-dihydroxy vitamin D3, restores autophagy in *P. gingivalis*-infected macrophages, lowering bacterial survival and increasing lysosomal function [76]. Rapamycin also activates *P. gingivalis*-infected myeloid dendritic cells [77]. Dental plaque anaerobic microbes produce butyrate, which promotes autophagy and caspase-independent oral keratinocyte cell death [78]. Other structural components stimulate autophagosome formation but limit autophagic flow, while periodontal metabolic byproducts may augment oral gingival autophagy. While the autophagosome helps *P. gingivalis* survive in the internal eukaryotic environment, some bacteria that enter early endosomes are exocytosed via the recycling endosome route. Certainly, gingival epithelial cells knocked down for recycling endosome marker RAB11, the small GTPase essential for interaction with exocyst constituents RAS like proto-oncogene A, RALA, or the exocyst complex constituents 2/3/84, EXOC2/3/84, and infected with *P. gingivalis*, showed decreased cfu. Plated internal content of infected gingival cells increases bacterial cfu, reducing exocytosis [7]. Since RALB assembles BECN1 components during starvation-induced autophagosome formation, *P. gingivalis* can grow and infect neighboring gingival cells via the exocyst channel [7]. Malfunctioning autophagy has been linked to periodontal disorders, worsening tissue destruction, and advancing diseases [79]. On the other hand, using drugs to control autophagy shows potential as a treatment approach for regulating periodontal inflammation and tissue damage. Oxidative stress-driven autophagy is a response to oral mucosa and periodontal tissue bacterial infection; therefore, increased ROS may not always cause periodontitis [80]. After microbial infections, neutrophils express increased ATG12 and LC3 to generate ROS [81]. N-acetylcysteine therapy of periodontitis-related neutrophils avoids ATG12 and LC3 gene expression, so ROS promote autophagy, possibly aiding cell survival [7]. This may reverse periodontitis-induced NFE2L2 downregulation. Since oral autophagy declines with age, oxidative stress worsens periodontal toxicity and bone loss from periodontitis.

Odontogenesis, often known as tooth development, refers to the biological process through which teeth are created from embryonic cells, undergo growth, and eventually emerge in the oral cavity [82]. The three main stages of odontogenesis are referred to as growth, calcification, and eruption. The human body undergoes its most extended period of development during this phase, which surpasses the growth duration of any other organ system. Odontogenesis involves autophagy to create energy and eliminate waste from enamel and oral epithelial cells [83]. Senescent dental pulp cells from older rats show increased LC3 and BECN1 immunohistochemical staining and reduced PPARg, a transcription factor needed for autophagy-related protein synthesis [84]. LPS stimulation depletes autophagy-related proteins, suggesting autophagy flux reduction causes LC3 and BECN1 accumulation [85]. CXCL12 increases LC3-II and ATG5-ATG12 complex levels and inhibits mTOR signaling in dental pulp stem cells from canine and human dental roots, favoring autophagy [7]. This model shows heavy ATG5-ATG12 complex and LC3 expression during tooth pulp development and regeneration, indicating autophagy [86]. Autophagy is essential for early tooth repair in hydrogen peroxide-treated mouse odontoblasts, as the LC3-II/LC3-I ratio, ATG12 levels, and ULK1-activating phosphorylations rise rapidly [7]. Simvastatin, an obesity-treating statin-family mTOR inhibitor, reduced tooth damage in a similar model [87]. Autophagy in periapical lesions increases odontoblast survival under adverse stimuli because chloroquine lowers pre-odontoblast cell line mDPC6T viability in LPS [88]. Since autophagy flux declines with age, dental repair ability decreases, emphasizing the need for novel polymers with higher repair and integration in older people [89].

A periapical lesion-associated eukaryotic pathogen, *Candida albicans* (*C. albicans*), requires autophagy for infection and pathogenicity [90]. The fungal homolog of PIK3C3, VPS34, binds with H^+^-ATPase’s Vma7 subunit to acidify vacuoles and activate autophagy lysosomes [91]. Nitrogen shortage kills VPS34-deficient *C. albicans*. Their survival is greatly reduced, demonstrating that *C. albicans* virulence requires fungal autophagy [92]. *C. albicans* autophagy increases after periapical lesions but decreases in the elderly [93]. Autophagy stimulation can restrict the overabundance of inflammation in periodontal tissue by suppressing the secretion of IL-1β, the development of NLRP3 inflammasome, and the buildup of ROS [94] (Figure 4).

Oral cancer develops when cells in the oral tissues undergo genetic mutations [95]. Typically, oral cancer originates in the squamous cells that form a thin layer on the surface of oral mucosa [96]. Minor alterations to the DNA of the squamous cells result in aberrant cellular growth [97]. Oral cancer presents a significant challenge due to its aggressive characteristics and elevated fatality rates. The relationship between autophagy and oral cancer is complex, as autophagy plays functions in suppressing and promoting tumors depending on the circumstances [98]. However, it can also help tumors survive when there is a lack of nutrients and oxygen. In oral carcinogenesis, mice injected subcutaneously with human tongue squamous cell carcinoma (TSCC) cells downregulated for BECN1 had larger and heavier tumor xenografts than controls [99]. In mice treated for 16 weeks with 4-nitroquinoline N-oxide, 4-NQO, a cigarette-smoke chemical, LC3, and SQSTM1/p62 levels rise as the oral mucosa becomes dysplastic and carcinogenic, showing that decreased autophagy causes malignant transformation [7]. These studies demonstrate autophagy suppression causes oral and non-oral cancer. Oral cancer progression should be monitored carefully or over time. Cal 27, a human OSCC cell line, treated with up to 2 mM melatonin converts LC3-I to LC3-II, lowers SQSTM1/p62, and enhances growth [100]. After 100 mg/kg melatonin, Cal 27 subcutaneous tumor xenograft mice lose weight, demonstrating that autophagy kills OSCC cells [101]. TFE3, an IGHM enhancer-binding transcription factor, upregulates autophagy genes such as Atg7 and Lamp1 [102]. Sepantronium bromide, a Survivin inhibitor, causes Cal 27 cell apoptosis and autophagy by increasing LC3 lipidation and SQSTM1/p62 degradation [7]. This model increases autophagy by lowering mTOR activity, as evidenced by decreased mTOR auto-activating phosphorylation at Ser-2448 and ribosomal protein S6, RPS6, phosphorylation at Ser-235 and Ser-236, and mTOR downstream [7]. Most intriguingly, 5 mg/kg sepantronium bromide therapy in tamoxifen-induced double knock-out mice for TGFBR1 and PTEN limits tumor growth and mTOR phosphorylation at Ser-2448 and SQSTM1/p62 immunohistochemistry [103]. Furthermore, the induction of autophagy during oral cancer progression slows tumor growth [104]. Utilizing autophagy pathways is a new strategy to improve the effectiveness of traditional cancer treatments and overcome resistance to therapy in oral malignancies.

In addition, systemic diseases such as diabetes mellitus may have a significant impact on oral health by affecting the dynamics of autophagy in oral tissues [105]. Changes in autophagic activity have been linked to the development of diabetic comorbidities, such as periodontal diseases and decreased wound healing [106]. A comprehensive understanding of the complex relationship between autophagy, overall health, and oral illnesses is crucial for creating comprehensive therapeutic approaches and enhancing patient outcomes [107]. Modulating autophagy has the potential of being therapeutically beneficial for several oral illnesses, providing new opportunities for preventing and treating these conditions. Pharmacological substances that specifically affect autophagy pathways such as rapamycin and chloroquine have shown effectiveness in experimental models of oral illnesses and have the potential for use in clinical settings [108]. Nevertheless, additional research is needed to fully understand the specific molecular mechanisms that regulate autophagy in relation to oral disorders. This will improve treatment approaches and reduce any potential negative consequences.

Moreover, human papillomavirus (HPV)-positive head and neck squamous cell carcinoma (HNSCC) exhibits unique clinical, genetic, and epidemiological characteristics in contrast to HPV-negative patients [109]. Nonetheless, the influence of HPV infection on autophagy in HNSCC remains inadequately investigated. It has been indicated that host cell autophagy is essential for facilitating cancer cell metabolism in a non-cell autonomous way, hence aiding tumor development [110,111]. In HPV16-positive HNSCC, autophagy is involved in multiple cancer-associated pathways. HPV16 E7 has been demonstrated to promote mitophagy and enable the autophagy-mediated degradation of STING (stimulator of interferon genes), a crucial component in immune responses [112]. This data suggests that HPV16 may regulate autophagy in a twofold fashion. HPV16 infection may suppress specific forms of autophagy, as seen by reduced LC3 flux, a hallmark of autophagic activity [113]. Conversely, HPV16 may facilitate specific types of autophagy, including mitophagy, to eliminate damaged mitochondria or induce the autophagic destruction of immune signaling proteins such as STING, thereby suppressing anti-tumor immune responses [114]. Additional examination of these processes may elucidate how HPV modifies autophagy to facilitate HNSCC advancement and immune evasion.

## 4. Recent Drug Target of Autophagy Modulation in Oral Diseases

Recent progress in comprehending the molecular underpinnings of autophagy has resulted in the discovery of possible pharmaceutical targets for regulating autophagy in oral disorders. This section explores the latest advancements in pharmacological targets for controlling autophagy in oral diseases. Specifically, it focuses on natural substances, synthesized pharmaceuticals, and medications that have been approved by the World Dental Federation (FDI).

### 4.1. Natural Compounds Targeting Autophagy in Oral Diseases

Due to varied chemical structures and pharmacological effects, natural substances have garnered significant interest as possible treatments for controlling autophagy. Various natural substances have been recognized for their capacity to regulate autophagy in oral disorders (Table 1). Curcumin, a polyphenol obtained from the rhizome of *Curcuma longa*, has undergone a thorough investigation due to its notable anti-inflammatory and anti-cancer characteristics [115]. Curcumin can stimulate autophagy in oral cancer cells by activating AMPK and suppressing the mTOR pathway [116]. Resveratrol, a natural polyphenol present in grapes, berries, and peanuts, has been identified to regulate autophagy in oral mucosal illnesses [117]. The effects of resveratrol are exerted through the activation of Sirtuin 1 (SIRT1) and AMPK pathways, resulting in the induction of autophagy and rejuvenation of cells [118]. Green tea catechins, specifically epigallocatechin gallate (EGCG), have been proven to improve the process of autophagy flux in oral epithelial cells [119]. EGCG induces autophagy by suppressing the PI3K/Akt/mTOR signaling pathway and enhancing the expression of autophagy-related genes [116]. Quercetin, an abundant flavonoid in fruits and vegetables, demonstrates strong anti-inflammatory and antioxidant characteristics [120]. Quercetin regulates the expression of autophagy-related proteins and removes reactive oxygen species (ROS), hence influencing autophagy in oral diseases [121]. Sulforaphane, obtained from cruciferous vegetables, triggers autophagy in oral mucosal illnesses by activating the Nrf2 pathway, thereby safeguarding cells against the effects of oxidative stress along with inflammation [122]. Berberine derived from the Berberis plant, stimulates autophagy in periodontitis by suppressing mTOR signaling, resulting in the elimination of germs and a decrease in inflammation of the gums [123]. Ellagic acid present in pomegranates induces autophagy in periodontitis via controlling mTOR signaling. This leads to the elimination of germs and a decrease in inflammation in the gums [124]. Ursolic acid, derived from apples and basil, stimulates autophagy in oral mucosal illnesses by triggering the AMPK pathway, thereby safeguarding cells from oxidative damage and inflammation [125]. Luteolin, a compound present in many vegetables and fruits, triggers autophagy in periodontitis by suppressing the mTOR pathway via AMPK stimulation. This leads to a decrease in periodontal inflammation as well as tissue harm [126]. Genistein, found in soybeans, induces autophagy in oral cancer cells by regulating the AMPK/mTOR pathway, hence suppressing tumor development and metastasis [121].

### 4.2. Synthetic Drugs Targeting Autophagy in Oral Diseases

Aside from naturally occurring substances, synthetic compounds are developed to target autophagy for the treatment of oral ailments. These medications frequently demonstrate greater specificity and efficacy, which makes them highly promising candidates for therapeutic application. Several synthetic medicines that specifically target autophagy in oral disorders are presented in Table 2. Chloroquine and its derivative hydroxychloroquine are substances that target lysosomes and hinder the process of autophagy by interfering with the normal functioning of lysosomes [127,128]. Chloroquine medications have been studied for their potential to improve the effectiveness of chemotherapy in treating oral cancer by blocking autophagic flow [129]. Rapamycin and its analogs, known as rapalogs, are inhibitors of the mTORC1 complex that stimulates autophagy by obstructing its activity [130]. These medicines have demonstrated encouraging outcomes in preclinical investigations for the management of oral mucosal diseases and periodontitis by enhancing the process of autophagic removal of damaged tissues. Metformin, a commonly given medication for diabetes, has been found to regulate autophagy in oral illnesses via activating AMPK [131]. It increases the process of autophagy and reduces inflammation in the cells that line the mouth, therefore slowing down the development of oral mucosal illnesses [132]. Statins, which are medications used to decrease cholesterol levels, have been found to stimulate autophagy in oral cancer cells by activating AMPK and blocking the mevalonate pathway [133]. These medications demonstrate anti-tumor properties and make oral cancer cells more susceptible to cell death caused by chemotherapy. Bafilomycin A1 is a medication used to treat oral vesiculobullous disorders. It works by inhibiting the fusion of autophagosomes with lysosomes [134]. LY294002 is a substance that hinders the activity of class III PI3K, which in turn prevents the process of autophagy in oral granulomatosis [135]. 3-Methyladenine is a substance that, when taken orally, can cause ulcers in the mouth. It functions by inhibiting class III PI3K, which in turn blocks the process of autophagy [136]. Vorinostat is an oral medication used to treat squamous cell carcinoma. It inhibits HDACs, which leads to the induction of autophagy [137]. Hydroxychloroquine is an oral medication that inhibits the fusion of autophagosomes with lysosomes, which is involved in the development of leukoplakia [138]. Temsirolimus is a substance that inhibits the mTOR pathway, which leads to the induction of autophagy, and is used to treat oral submucous fibrosis [139].

### 4.3. FDI-Approved Drugs Targeting Autophagy in Oral Diseases

The FDI evaluates the safety and effectiveness of medications for managing oral diseases before approval. Although there are currently no medications specifically licensed for targeting autophagy in oral disorders, numerous FDI-approved medications utilized in the management of oral diseases can modulate autophagy [140] (Table 3). For example, chlorhexidine is a wide-ranging antiseptic that is frequently employed to prevent and treat periodontitis [141]. There have been reports indicating that it can stimulate autophagy in oral epithelial cells by activating AMPK and suppressing mTOR signaling [142]. Doxycycline, a kind of tetracycline antibiotic, is commonly used as an additional treatment for periodontitis because it has anti-inflammatory properties and can block matrix metalloproteinases [143]. Furthermore, it can stimulate autophagy in periodontal ligament cells, hence promoting tissue regeneration and maintaining periodontal health [144]. Fluoride, a mineral present in dental products like toothpaste and mouthwash, is crucial for the prevention of dental cavities [145]. Recent research indicates that fluoride may regulate autophagy in oral epithelial cells [146], while the precise mechanism is yet unknown.

## 5. Limitations and Future Perspectives of Autophagy and Oral Health

The significance of autophagy in oral health has received considerable attention in recent years, due to its involvement in several oral disorders such as periodontitis, oral cancer, and oral infections. Although autophagy has a therapeutic potential for dental health, its implementation is restricted by various issues, which are discussed in this section. In addition, the limitations and potential future developments of autophagy in relation to dental health are discussed. The complexity of the oral environment and variability of oral disorders pose a significant challenge when investigating autophagy in oral health. Various oral disorders demonstrate unique autophagic reactions, posing a challenge for developing universal treatment approaches. The regulation of autophagy entails an intricate network of signaling channels, indicating the presence of sophisticated regulatory mechanisms [157]. Imbalance in these pathways may lead to either excessive or inadequate autophagy, which plays a role in the development of oral disorders. The oral cavity contains a varied microbial population that engages in interactions with host cells [158]. The precise mechanisms of autophagy that influence the interactions between the host and the microbiota, and oral health, are not well comprehended. The level of autophagic activity differs across various oral tissues, including the gingiva, periodontal ligament, and salivary glands [159]. Gaining a comprehensive understanding of the dynamics of autophagy in different tissues is crucial for developing precise and focused therapeutic interventions. Currently, there is a scarcity of diagnostic instruments accessible for evaluating autophagic activity in clinical environments. It is essential to identify dependable biomarkers of autophagy in oral tissues for the early diagnosis of diseases and to monitor the effectiveness of treatments. Delivering medications that modify autophagy in oral tissues is difficult because of the intricate structure of the mouth and the presence of saliva, which can impact the availability and effectiveness of the drugs. Autophagy modulators sometimes have off-target effects, which can result in unexpected repercussions in oral tissues [160]. Precise targeting of autophagy pathways is essential to minimize any negative consequences. Furthermore, the ethical implications of utilizing autophagy-modulating treatments, specifically in susceptible demographics including children and pregnant women, must be acknowledged.

Future research on genetic markers associated with autophagy may yield significant insights into disease prognosis in oral cancer, particularly for patients undergoing chemotherapy. Studies are needed on autophagy-related gene polymorphisms and cancer prognosis, notably the ATG5 rs473543 polymorphism and disease-free survival in breast cancer [161]. Investigating these markers may not only improve our comprehension of autophagy’s function in tumor advancement and therapy efficacy, but also facilitate the development of tailored therapeutic approaches. Determining distinct autophagy-related genetic profiles may facilitate the prediction of patient responses to chemotherapies and assess the possibility of drug resistance, thereby establishing a basis for more effective and personalized treatment strategies. Additionally, vitamin D receptor (VDR) polymorphisms, such as FokI, BsmI, TaqI, ApaI, and Cdx2, play important roles in regulating bone health and immune responses, which are essential for maintaining oral homeostasis. Variations in these genes can influence susceptibility to oral diseases, particularly periodontal conditions and bone-related disorders. These polymorphisms play a significant role in oral health and homeostasis, affecting processes such as bone metabolism, immune responses, and the regulation of calcium and phosphate levels in the oral cavity [162].

Targeted therapeutics can be developed for specific oral disorders by advancing our understanding of autophagy regulation. Customizing treatment approaches according to the autophagic condition of specific patients shows potential for personalized medicine. Combination therapies involve using autophagy modulators in conjunction with conventional treatments, like antibiotics and chemotherapy, to improve the effectiveness of treatment and decrease the development of drug resistance in oral illnesses. Specific dietary components have demonstrated the ability to regulate autophagy and enhance dental health [163]. Investigating the capacity of nutraceuticals and dietary treatments to control autophagy may provide innovative methods for preventing and treating diseases. Microbiota-targeted approaches involve studying the relationship between autophagy and the oral microbiota to develop medicines that specifically target the microbiota for treating oral disorders [164]. Probiotics and prebiotics that support advantageous microbial communities can improve autophagic function and oral health [165]. The identification of dependable biomarkers for autophagy in oral tissues can aid in the early diagnosis of diseases and the monitoring of therapy effectiveness. Omics technologies, including genomes and proteomics, offer the potential for identifying new autophagy biomarkers [166]. The advancement of non-invasive imaging techniques, such as saliva-based assays and oral microbiota analysis, can offer valuable information about autophagic activity in oral tissues without requiring invasive procedures. Utilizing nanoparticles as carriers, drug delivery systems based on nanotechnology have the capability to precisely administer autophagy modulators to oral tissues [167], hence enhancing the effectiveness of drugs and reducing unintended side effects. Raising public awareness on the significance of autophagy in oral health and the potential advantages of autophagy-focused treatments can encourage patients to embrace preventive measures and comply with treatment plans.

## 6. Conclusions

Autophagy has a crucial role in controlling inflammation and maintaining oral health. Oral diseases such as periodontitis, gingivitis, and oral cancer are characterized by dysregulated inflammation. Autophagy regulates inflammatory reactions by eliminating impaired cellular components and inhibiting the secretion of pro-inflammatory cytokines [168]. Autophagy has a role in preserving periodontal health and preventing excessive tissue damage from long-term inflammation. In addition, autophagy plays a role in controlling the activity of immune cells, such as macrophages, dendritic cells, and T cells [169], and eliminating intracellular pathogens such as bacteria and viruses [170], Oral epithelial cells are consistently subjected to various environmental stresses such as microbial toxins, reactive oxygen species, and mechanical damage [171]. Autophagy aids cellular stress management by eliminating impaired organelles and aggregated proteins, therefore averting cellular malfunction and facilitating tissue restoration. The disruption of autophagy has been linked to the development of several oral diseases including periodontal diseases and oral cancer [172]. Comprehending the molecular processes that cause autophagy malfunction in certain circumstances may lead to the creation of new treatment approaches that specifically target autophagy pathways. Additional investigation into the intricate relationship between autophagy and oral health will not only enhance our comprehension of oral pathophysiology, but also present novel prospects for the prevention and management of oral diseases.

## Figures and Tables

**Figure 1 biomedicines-12-02645-f001:**
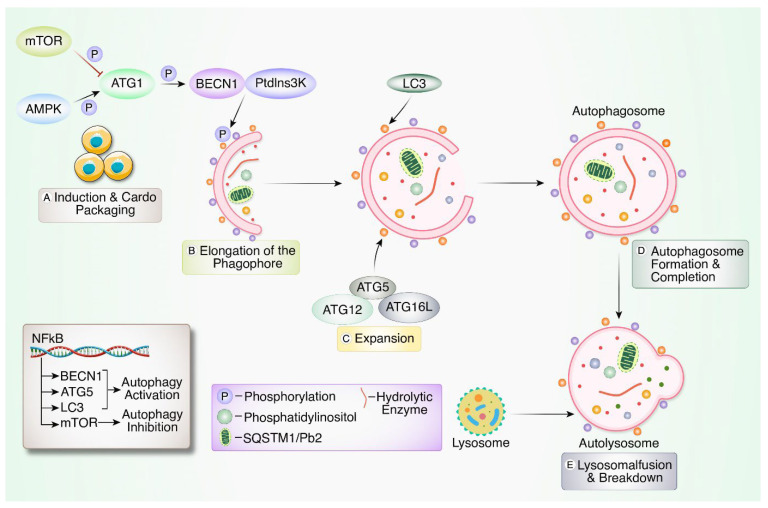
Mechanistic steps of autophagy. Autophagy is initiated by recognized signaling pathways like AMP-activated protein kinase (AMPK) activation or mTOR inhibition, phosphorylating ATG1 and Beclin 1 (BECN1) (A). The autophagosomes elongate (B) and incorporate LC3 molecules with assistance from ATG5, while components targeted for degradation reach the autophagosome via receptors like SQSTM1/p62, binding polyubiquitinated proteins (C). Following autophagosome formation (D), it fuses with a lysosome containing hydrolytic enzymes, leading to material degradation (E). The NFκB pathway transcriptionally upregulates autophagy proteins (e.g., BECN1, LC3, and ATG5), promoting autophagy.

**Figure 2 biomedicines-12-02645-f002:**
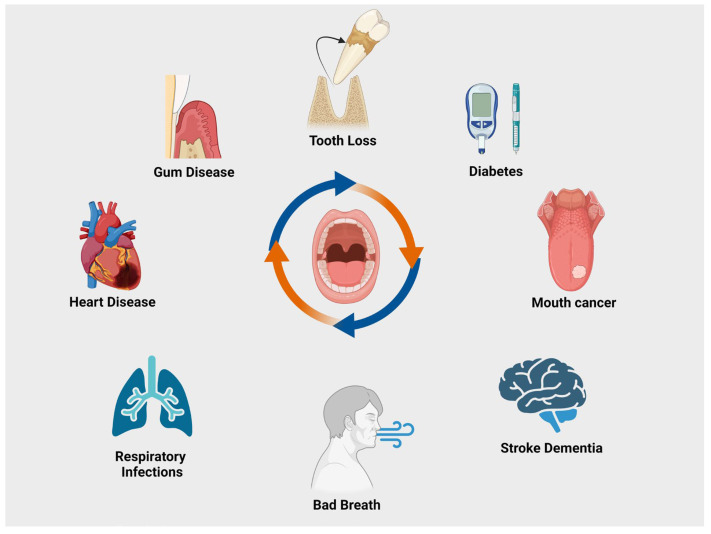
Associations with oral health and systemic conditions. Oral health is associated with numerous oral conditions and systemic diseases, including gum disease, heart disease, diabetes, respiratory infections, stroke, and dementia.

**Figure 3 biomedicines-12-02645-f003:**
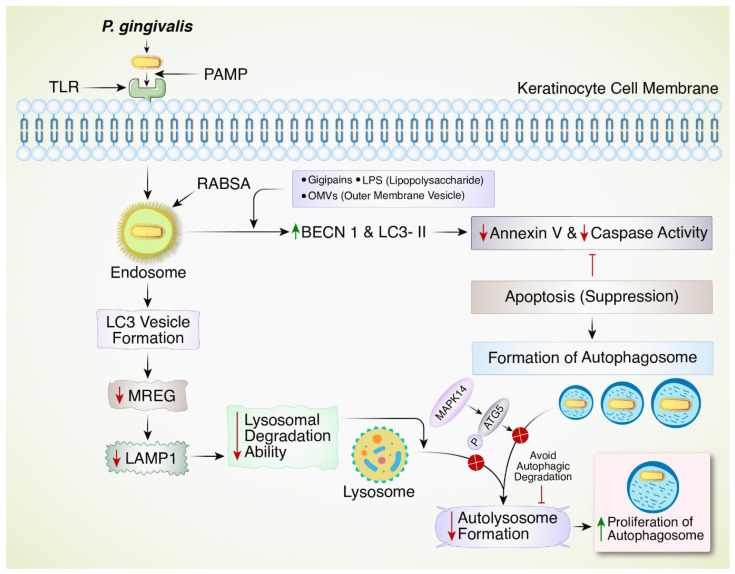
Autophagic process of *P. gingivalis* pathogenicity in antigen-presenting cells. *P. gingivalis* interacts with PAMP-TLR on the cell membrane, entering antigen-presenting cells in the oral cavity, and co-localizing with RAB5A-positive vehicles in early endosome formation. *P. gingivalis* virulence factors (gingipains, LPS, OMVs) increase BECN1 and LC3-II that decrease annexin V and caspase activity, thus suppressing apoptosis and forming autophagosomes. In an alternative pathway, RAB5A-positive vesicles increase LC3 vesicle formation that reduces MREG and LAMP1, lowering lysosomal activity and autolysosome formation. Additionally, MAPK14 phosphorylating ATGs further reduces autolysosome formation. Both pathways increase the proliferation of autophagosomes.

**Figure 4 biomedicines-12-02645-f004:**
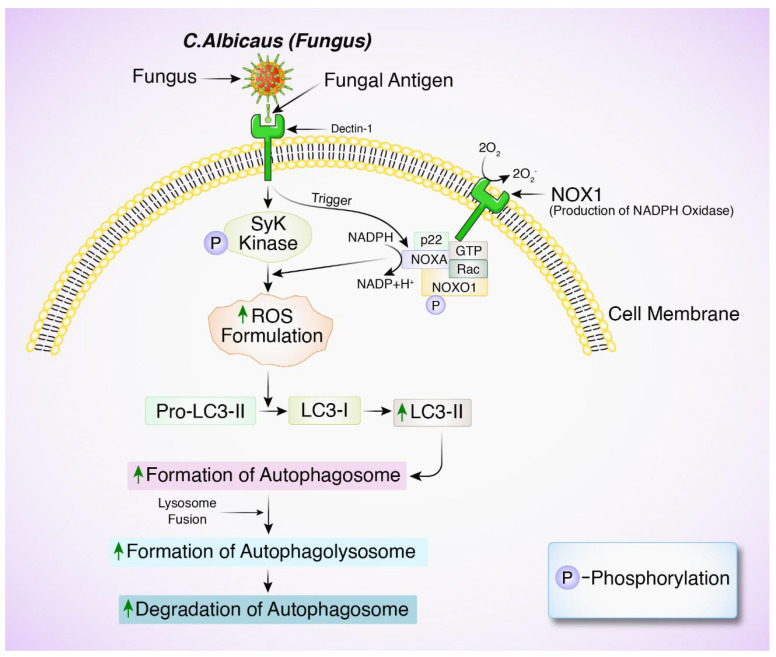
The role of autophagy in *C. albicans* infection in oral keratinocytes. Fungal antigen recognized by extracellular Dectin-1 receptor, triggers phosphorylation of tyrosine residue by Src kinases, recruiting Syk (spleen tyrosine kinase). This activation leads to the recruitment of NADPH oxidase via NOX1 oxidation and NOXO1 phosphorylation, resulting in the release of antimicrobial reactive oxygen species (ROS) into the phagosome. ROS generation triggers LC3 processing and recruitment of LC3-II, thus facilitating phagosome formation, maturation, and fusion with lysosomes, eventually leading to autophagosomal degradation.

**Table 1 biomedicines-12-02645-t001:** Natural compounds targeting autophagy in oral diseases.

Compound	Source	Mechanism of Action	Oral Disease Targeted	References
Curcumin	Turmeric	Activation of AMPK, inhibition of mTOR pathway	Oral mucosal disorders	[115]
Resveratrol	Grapes, berries	Activation of SIRT1, AMPK pathways	Oral cancer, periodontitis	[117]
Epigallocatechin gallate (EGCG)	Green tea	Inhibition of PI3K/Akt/mTOR pathway	Periodontitis	[119]
Quercetin	Fruits, vegetables	Regulation of autophagy-related proteins, ROS scavenging	Oral cancer	[120]
Sulforaphane	Cruciferous vegetables	Induces autophagy through Nrf2 activation	Oral mucosal disorders	[122]
Berberine	Berberis plant	Promotes autophagy by inhibiting mTOR signaling	Periodontitis	[123]
Ellagic acid	Pomegranates	Stimulates autophagy via regulation of mTOR signaling	Periodontitis	[124]
Ursolic acid	Apples, basil	Promotes autophagy by activating AMPK pathway	Oral mucosal disorders	[125]
Luteolin	Vegetables, fruits	Activates autophagy through AMPK/mTOR inhibition	Periodontitis	[126]
Genistein	Soybeans	Stimulates autophagy via AMPK/mTOR pathway	Oral cancer	[121]

**Table 2 biomedicines-12-02645-t002:** Synthetic drugs targeting autophagy in oral diseases.

Synthetic Drug	Mechanism of Action	Targeted Oral Disease	References
Chloroquine	Inhibition of lysosomal function	Oral Cancer	[129]
Rapamycin/Rapalogs	Inhibition of mTORC1 complex	Periodontal Disease	[130]
Metformin	AMPK activation	Oral Lichen Planus	[132]
Statins	AMPK activation, inhibition of mevalonate pathway	Oral Cancer	[133]
Bafilomycin A1	Blocks autophagosome–lysosome fusion	Oral Vesiculobullous Disorders	[134]
LY294002	Inhibits class III PI3K, blocking autophagy	Oral Granulomatosis	[135]
3-Methyladenine	Inhibits class III PI3K, blocking autophagy	Oral Ulcers	[136]
Vorinostat	Inhibits HDACs, inducing autophagy	Oral Squamous Cell Carcinoma	[137]
Hydroxychloroquine	Inhibits autophagosome–lysosome fusion	Oral Leukoplakia	[138]
Temsirolimus	Inhibits mTOR pathway, inducing autophagy	Oral Submucous Fibrosis	[139]

**Table 3 biomedicines-12-02645-t003:** Pharmaceuticals authorized by the FDI to specifically target autophagy in oral disorders.

FDI Drug Name	Mode of Action in Autophagy	Oral Disease	References
Rapamycin	Inhibits mTOR, thereby inducing autophagy	Oral squamous cell carcinoma	[147]
Metformin	Activates AMPK, promoting autophagy	Periodontitis	[148]
Aspirin	Induces autophagy through inhibition of mTOR	Oral squamous cell carcinoma	[149]
Everolimus	mTOR inhibitor, enhances autophagy	Head and neck cancers	[150]
Chloroquine	Inhibits lysosomal function, impairing autophagy flux	Oral lichen planus	[151]
Doxycycline	Induces autophagy by activating AMPK	Periodontitis and oral infections	[152]
Simvastatin	Activates AMPK, inducing autophagy	Oral cancer and periodontitis	[153]
Tamoxifen	Inhibits mTOR signaling, promotes autophagy	Oral cancer	[154]
Curcumin	Induces autophagy through AMPK and mTOR regulation	Oral potentially malignant disorders	[155]
Resveratrol	Activates SIRT1, promoting autophagy	Periodontitis and oral cancers	[156]

## Data Availability

All necessary data generated or analyzed during this study are presented in this article and additional data could be available from the corresponding author upon request.

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
