# Peer review of "Advancements in Autophagy Modulation for the Management of Oral Disease: A Focus on Drug Targets and Therapeutics"

_biomedicines, 2024, doi:10.3390/biomedicines12112645_

Round 1

Reviewer 1 Report

Comments and Suggestions for Authors

This article provides an insightful examination of autophagy as a fundamental cellular process involved in maintaining oral health and offers a perspective on the molecular mechanisms that influence the progression of oral diseases. The authors effectively highlight the significance of autophagy in cellular recycling and its impact on metabolic pathways, differentiation, and cell survival. Recognizing oral health as integral to overall well-being, the authors emphasize the importance of understanding the intricate interactions between autophagy and oral health for therapeutic innovation.

The discussion on the influence of autophagy on both hard and soft tissues in the oral cavity and its interplay with cellular aging is particularly relevant. Aging is often associated with reduced autophagic activity, which aligns with increased vulnerability to oral disorders such as periodontitis, oral cancer, and periapical lesions. This point underscores the potential for targeting autophagy as a therapeutic strategy in age-related oral health deterioration.

The authors also address the gap in the literature on the molecular mechanisms linking autophagy to oral health, signaling a promising direction for further research. By connecting autophagy levels to the prevention and treatment of specific oral diseases—such as oral cancer, periapical lesions, and candidiasis—the review identifies autophagy as a potential target for novel therapies.

In conclusion, this review contributes to the growing body of knowledge on autophagy's role in oral health and advocates for more research to elucidate the specific molecular pathways involved. This enhanced understanding could be transformative for preventive and therapeutic strategies, ultimately improving outcomes for various oral diseases.

Author Response

Reviewer 1

This article provides an insightful examination of autophagy as a fundamental cellular process involved in maintaining oral health and offers a perspective on the molecular mechanisms that influence the progression of oral diseases. The authors effectively highlight the significance of autophagy in cellular recycling and its impact on metabolic pathways, differentiation, and cell survival. Recognizing oral health as integral to overall well-being, the authors emphasize the importance of understanding the intricate interactions between autophagy and oral health for therapeutic innovation.

>>Response: We appreciate your thoughtful and encouraging comments on our manuscript. We appreciate your recognition of our efforts to understand autophagy's vital function in dental health. As you said, understanding how autophagy affects recycling, metabolic control, differentiation, and cell survival can lead to new oral disease treatments. We agree that oral health is important to general well-being and want to study autophagy and specific oral disease pathological pathways.

The discussion on the influence of autophagy on both hard and soft tissues in the oral cavity and its interplay with cellular aging is particularly relevant. Aging is often associated with reduced autophagic activity, which aligns with increased vulnerability to oral disorders such as periodontitis, oral cancer, and periapical lesions. This point underscores the potential for targeting autophagy as a therapeutic strategy in age-related oral health deterioration.

>>Response: Your kind comments on our manuscript are appreciated. We're delighted you found the debate on autophagy's impact on oral hard and soft tissues and cellular aging relevant. Aging diminishes autophagic activity, which increases the risk of periodontitis, oral cancer, and periapical diseases. This supports our focus on autophagy as a treatment target for age-related dental health decline. After your remarks, we are considering how autophagy modulation strategies may assist age-related oral problems. We'll gladly incorporate your more suggestions into our improvements. Thank you again for your positive and constructive remarks.

The authors also address the gap in the literature on the molecular mechanisms linking autophagy to oral health, signaling a promising direction for further research. By connecting autophagy levels to the prevention and treatment of specific oral diseases—such as oral cancer, periapical lesions, and candidiasis—the review identifies autophagy as a potential target for novel therapies.

>>Response: We appreciate your recognition of our efforts to fill the literature gap in autophagy and oral health molecular pathways. We wanted to show how autophagy activity affects oral cancer, periapical lesions, and candidiasis outcomes and treatments. We want to inspire future research on how modifying autophagy pathways can prevent or treat oral illnesses by targeting autophagy. We value your feedback and welcome recommendations to deepen and clarify our work.

In conclusion, this review contributes to the growing body of knowledge on autophagy's role in oral health and advocates for more research to elucidate the specific molecular pathways involved. This enhanced understanding could be transformative for preventive and therapeutic strategies, ultimately improving outcomes for various oral diseases.

>>Response: Thanks for emphasizing our review's conclusion. We're glad you liked the debate on autophagy's involvement in oral health and the demand for more research on specific cellular pathways. We agree that knowing these pathways could improve oral disease prevention and treatment. We are happy to add details or clarity to the article if needed. Thank you again for your thoughtful feedback and encouraging perspective on our work's potential impact.

Reviewer 2 Report

Comments and Suggestions for Authors

The manuscript is overall ok, however, a few points need to be added

1 Autophagy could be a double edge weapon, for instance inhibiting intestinal epithelial autophagy through intestinal flora could improve colon rectal cancer patients’ responses to chemotherapy and alter outcomes (Yu T, Guo F, Yu Y, Sun T, Ma D, Han J, et al.. Fusobacterium nucleatum promotes chemoresistance to colorectal cancer by modulating autophagy. Cell (2017) 170(3):548–63.e16), while another study confirmed that in ovarian cancer, the resistance of the cytotoxic drug paclitaxel has been attributed to autophagy induction (Zhang SF, Wang XY, Fu ZQ, Peng QH, Zhang JY, Ye F, et al.. Txndc17 promotes paclitaxel resistance via inducing autophagy in ovarian cancer. Autophagy (2015) 11(2):225–38). I suggest the author add this topic, which may give a wider perspective on this specific matter.

2 Recently, the relationships between autophagy-related gene variants, polymorphisms, and cancer prognosis have been evaluated. For example, the association between rs473543 in ATG5 and disease-free survival (DFS) of breast cancer patients undergoing chemotherapy was reported (Li M, Ma F, Wang J, Li Q, Zhang P, Yuan P, et al.. Genetic polymorphisms of autophagy-related gene 5 (atg5) rs473543 predict different disease-free survivals of triple-negative breast cancer patients receiving anthracycline- and/or taxane-based adjuvant chemotherapy. Chin J Cancer (2018) 37(1):4) 

3 I would suggest adding something related to VDR polymorphisms that play a crucial role in oral homeostasis.

4 We should consider the possible effect on autophagy by some virus or bacteria. Eg. Human papillomavirus (HPV)-positive head and neck squamous cell carcinoma (HNSCC) displays distinct epidemiological, clinical, and molecular characteristics compared to its negative counterpart. However, the influence of HPV infection on autophagy in HNSCC has received less attention and has not been previously reviewed. Again, host cell autophagy can support cancer cell metabolism in a non-cell autonomous manner (Sousa et al. 2016; Katheder et al. 2017). Moreover, HPV16 E7 was reported to induce lethal mitophagy as well as autophagy-dependent degradation of STING in HNSCC cells. It is a conceivable possibility that HPV16 infection limits some types of autophagy (as reflected by decreased LC3 flux), whilst increasing certain selective types of autophagy (e.g., of mitochondria and STING).

Comments on the Quality of English Language

Minor English editing is needed

Author Response

Reviewer 2

The manuscript is overall ok, however, a few points need to be added

  1. Autophagy could be a double edge weapon, for instance inhibiting intestinal epithelial autophagy through intestinal flora could improve colon rectal cancer patients’ responses to chemotherapy and alter outcomes (Yu T, Guo F, Yu Y, Sun T, Ma D, Han J, et al.. Fusobacterium nucleatum promotes chemoresistance to colorectal cancer by modulating autophagy. Cell (2017) 170(3):548–63.e16), while another study confirmed that in ovarian cancer, the resistance of the cytotoxic drug paclitaxel has been attributed to autophagy induction (Zhang SF, Wang XY, Fu ZQ, Peng QH, Zhang JY, Ye F, et al.. Txndc17 promotes paclitaxel resistance via inducing autophagy in ovarian cancer. Autophagy (2015) 11(2):225–38). I suggest the author add this topic, which may give a wider perspective on this specific matter.

>>Response: Your feedback and proposal to expand on autophagy's dual involvement in cancer therapy resistance are appreciated. Autophagy can promote and inhibit tumors, as we concur. The effects of autophagy on chemoresistance in different cancer types discussed. Yu et al. (2017) showed that inhibiting autophagy improves colorectal cancer outcomes with Fusobacterium nucleatum, and Zhang et al. (2015) showed that inducing autophagy causes paclitaxel resistance in ovarian cancer. We added this information section “2. Relationship between autophagy and oral health” page 4 line 163-175.

  1. Recently, the relationships between autophagy-related gene variants, polymorphisms, and cancer prognosis have been evaluated. For example, the association between rs473543 in ATG5 and disease-free survival (DFS) of breast cancer patients undergoing chemotherapy was reported (Li M, Ma F, Wang J, Li Q, Zhang P, Yuan P, et al.. Genetic polymorphisms of autophagy-related gene 5 (atg5) rs473543 predict different disease-free survivals of triple-negative breast cancer patients receiving anthracycline- and/or taxane-based adjuvant chemotherapy. Chin J Cancer (2018) 37(1):4) 

>>Response: Thank you for mentioning current studies on autophagy-related gene polymorphisms and cancer prognosis, notably the ATG5 rs473543 polymorphism and disease-free survival in breast cancer. Our work focuses on autophagy in oral disease and health, although genetic differences in autophagy-related genes like ATG5 may affect oral cancer outcomes. Researchers could examine similar genetic markers to determine their effects on autophagy and disease prognosis in oral cancer patients, particularly those receiving chemotherapy. We appreciate your suggestion and add a brief discussion on this growing field to underscore autophagy gene variations' importance in oral health research with that reference in section “5. Limitations and Future Perspectives of autophagy and oral health” page 14 line 531-539.

  1. I would suggest adding something related to VDR polymorphisms that play a crucial role in oral homeostasis.

>>Response: I appreciate the suggestion.  We address how VDR (Vitamin D Receptor) polymorphisms affect oral homeostasis, bone metabolism, immunological function, and oral tissue integrity. This strengthen the review by including oral health genetics with the reference in section “5. Limitations and Future Perspectives of autophagy and oral health” page 14 line 540-546.

  1. We should consider the possible effect on autophagy by some virus or bacteria. Eg. Human papillomavirus (HPV)-positive head and neck squamous cell carcinoma (HNSCC) displays distinct epidemiological, clinical, and molecular characteristics compared to its negative counterpart. However, the influence of HPV infection on autophagy in HNSCC has received less attention and has not been previously reviewed. Again, host cell autophagy can support cancer cell metabolism in a non-cell autonomous manner (Sousa et al. 2016; Katheder et al. 2017). Moreover, HPV16 E7 was reported to induce lethal mitophagy as well as autophagy-dependent degradation of STING in HNSCC cells. It is a conceivable possibility that HPV16 infection limits some types of autophagy (as reflected by decreased LC3 flux), whilst increasing certain selective types of autophagy (e.g., of mitochondria and STING).

>>Response: Thank you for the helpful suggestion. Autophagy in cancer, especially HPV-positive HNSCC, is understudied. Your point is well accepted, although HPV infection's effect on autophagy in this cancer subtype is less thoroughly explored. As you noted, HPV infection, particularly HPV16, alters cellular processes including autophagy, which may affect tumor growth and treatment response. We include these findings in the review and discuss how HPV infection modifies autophagic mechanisms in HNSCC to provide a complete and we added in the last paragraph of section “3. Interplay between autophagy and oral diseases” page 10 line 395-410 with recent references.

Minor English editing is needed

>>Response: I appreciate your suggestion. We thoroughly review English to increase readability, coherence, and clarity. This includes improving sentence structure, vocabulary, and minor grammatical errors and changing marked by BLUE color the entire manuscript. Please indicate any areas that need attention. We guarantee linguistic quality that meets your journal's high requirements.

Round 2

Reviewer 2 Report

Comments and Suggestions for Authors

The paper can be accepted